# Research Progress on Spike-Dependent SARS-CoV-2 Fusion Inhibitors and Small Molecules Targeting the S2 Subunit of Spike

**DOI:** 10.3390/v16050712

**Published:** 2024-04-30

**Authors:** Matthew R. Freidel, Roger S. Armen

**Affiliations:** Department of Pharmaceutical Sciences, College of Pharmacy, Thomas Jefferson University, 901 Walnut St. Suite 918, Philadelphia, PA 19170, USA; freidel36@gmail.com

**Keywords:** arbidol, toremifene, entry inhibitor, fusion inhibitor, Spike dependent, SARS-CoV-2, S2 segment, S2 subunit, surface plasmon resonance, molecular docking

## Abstract

Since the beginning of the COVID-19 pandemic, extensive drug repurposing efforts have sought to identify small-molecule antivirals with various mechanisms of action. Here, we aim to review research progress on small-molecule viral entry and fusion inhibitors that directly bind to the SARS-CoV-2 Spike protein. Early in the pandemic, numerous small molecules were identified in drug repurposing screens and reported to be effective in in vitro SARS-CoV-2 viral entry or fusion inhibitors. However, given minimal experimental information regarding the exact location of small-molecule binding sites on Spike, it was unclear what the specific mechanism of action was or where the exact binding sites were on Spike for some inhibitor candidates. The work of countless researchers has yielded great progress, with the identification of many viral entry inhibitors that target elements on the S1 receptor-binding domain (RBD) or N-terminal domain (NTD) and disrupt the S1 receptor-binding function. In this review, we will also focus on highlighting fusion inhibitors that target inhibition of the S2 fusion function, either by disrupting the formation of the postfusion S2 conformation or alternatively by stabilizing structural elements of the prefusion S2 conformation to prevent conformational changes associated with S2 function. We highlight experimentally validated binding sites on the S1/S2 interface and on the S2 subunit. While most substitutions to the Spike protein to date in variants of concern (VOCs) have been localized to the S1 subunit, the S2 subunit sequence is more conserved, with only a few observed substitutions in proximity to S2 binding sites. Several recent small molecules targeting S2 have been shown to have robust activity over recent VOC mutant strains and/or greater broad-spectrum antiviral activity for other more distantly related coronaviruses.

## 1. Introduction

The world has made significant strides since the emergence of the novel coronavirus in 2019 [1,2]. The following years saw collective efforts throughout the world as scientists and health policymakers initially attempted to contain the spread of the disease. As time wore on, it has become apparent that COVID-19—and the virus that causes it, SARS-CoV-2—will remain with us. Additionally, we have had to contend with new variants that can evade current treatments and vaccines, necessitating further resources and effort to keep up with evolution [3].

While we have come a long way in our clinical understanding of the disease and how to treat it, the cost of the pandemic on human lives and economic output has highlighted the need to be prepared for future pandemics. We have also seen that there will not likely be a one-size-fits-all approach. While immunoglobulin-based therapeutic approaches have been beneficial for outpatient treatment [4], in patients at higher risk for severe disease [5,6], as well as in patients hospitalized with severe disease [7,8,9,10], these therapies have limitations [11,12,13,14]. They require a great deal of invested capital and healthcare infrastructure that may not be available to rural communities or developing parts of the world and may otherwise be prohibitively expensive. Small molecules can overcome many of these hurdles, and when inhibitors are repurposed from existing drugs, the costs (and, therefore, a large barrier to access) are substantially reduced. However, to optimize the efficacy of small-molecule inhibitors, we need to look outside of some of the current strategies for small-molecule drugs. In some of our earlier research, we were attracted to non-structural protein 13, the helicase [15], for its highly conserved nature among coronaviruses. The same concept of targeting a highly conserved structural element would also lead us to focus our attention on the S2 segment of the Spike protein (Figure 1) found on the surface of the viral envelope of SARS-CoV-2. This region, which consists of 585 amino acids, makes up its constituent fusion peptide and two heptad repeats (HR1 and HR2) [2] and offers an alternative target for small-molecule inhibitors over the S1 receptor-binding domain (RBD) that is currently the primary focus of treatment.

## 2. The S2 Subunit Sequence Has Remained Very Conserved in Variants (VOCs)

During the summer of 2021, there was increasing concern regarding the emergence of the Omicron variant. This variant boasted enhanced transmissibility relative to earlier variants, which was the result of numerous substitutions to the Spike protein, specifically the S1 segment, the region that is responsible for binding with the angiotensin-converting enzyme 2 (ACE2) receptor on host cells [16,17]. The S1 segment is highly subject to substitution, and the majority of observed substitutions to Spike observed to date in variants of concern (VOCs) have been found on S1. Indeed, some immunoglobulin treatments themselves put selective pressure on variants, favoring those with altered S1 segments [18,19]. This has resulted in some of the immunoglobulin-based therapies and earlier vaccines losing their efficacy against some variants [16,17]. Thankfully, the development of resistance is not insurmountable. If the S1 segment is the tip of the spear, the S2 segment is the shaft, which is highly conserved due to its structure being crucial for function. Indeed, the S2 segment is not only conserved among SARS-CoV-2 variants but among other coronaviruses such as Middle East Acute Respiratory Syndrome (MERS), SARS-CoV-1, and even among zoonotic sarbecoviruses [20,21,22]. It is likely that this is due to the crucial role that structural rearrangement of S2 in coronaviruses has in stabilizing the postfusion conformation, allowing the delivery of viral RNA into the host cell.

With respect to specific targetable elements of the S2 subunit, the fusion peptide, stem helix, and the heptad repeat 1-heptad repeat 2 bundle (HR1-HR2 bundle), there exists little variation between different sarbecoviruses [20,21,22]. In a recent sequence-structure analysis of the substitutions from the Omicron variant and several related sublineages (BA.1, BA.2, and BA.3) [23], substitutions were observed at 31 sequence positions (84%) on the S1 subunit, while substitutions were only observed at 6 sequence positions (16%) on the S2 subunit. Three substitutions, Q954H, N967K, and L981F, are found on the HR1 motif, leaving the CH, CD, SH, and HR2 motifs entirely conserved among these lineages. The fusion peptide is remarkably conserved, which initiates the fusion event by inserting itself into the target cell membrane. The HR1-HR2 bundle specifically has few observed substitutions, while a recent functional mutation study demonstrated that the specific HR1 motif mutation L981F reduced fusion activity as a single point mutation [24], where allosteric effects of other mutations compensate for this effect in Omicron [24]. We outline below that small molecule inhibitors that disrupt the function of the postfusion 6HB structure are found to exhibit broad-spectrum activity against a range of coronavirus strains. While protein-based immunoglobulin treatments are not the focus of this review, a recent article by Guo et al. reviews and highlights S2-specific epitopes that are targeted by some new second-generation immunoglobulins, resulting in effective neutralization and a wider spectrum of activity [25]. Rather, we focus on small molecules and highlight where they putatively bind to the Spike protein to inhibit Spike function.

## 3. Conformational Changes and S2 Subunit Function

Upon binding with the ACE2 receptor, the S1 subunit will dissociate from the S2 at the S1/S2 junction [26,27]. This is mediated by the endopeptidase furin. This will then expose the S2′ site, which is then cleaved by the host’s transmembrane protease serine 2 via the plasma pathway, though it may also be mediated by lysosomal cathepsins during virion endocytosis [28,29]. This cleavage releases the structural constraints on the fusion peptide region and allows the flexibility for fusion to occur [30]. As SARS-CoV-2 uses the ACE2 receptor for cell entry, following receptor binding, there are two possible pathways for cell fusion: (1) membrane fusion of viral membrane and host membrane, or (2) membrane fusion of an infected cell and uninfected cells to form syncytium and facilitate cell-to-cell infection [31,32]. If the polybasic furin site exists on the sequence, as it does for SARS-CoV-2, but not for SARS-CoV, then the furin cleavage results in much greater cell-to-cell membrane fusion and formation of syncytia [31,32,33]. SARS-CoV, lacking the furin cleavage site, results in much less syncytia formation and, predominantly, cell entry that is dependent on clathrin-mediated endocytosis and the endosomal pathway. As the SARS-CoV-2 Spike protein is much more effective at cell-to-cell membrane fusion and formation of syncytia than SARS-CoV, this also highlights the concept of targeting the S2 subunit fusion function in SARS-CoV-2 infection, as this is one of several important differences that underly differences in disease pathogenesis and severity. Interestingly, some drug repurposing screens have identified entry or fusion inhibitors that are able to inhibit Spike-mediated syncytia formation [34], including Niclosamide, Clofazimine, and Salinomycin. While much remains elusive as to how some specific inhibitors act to prevent cell entry or membrane fusion, other peptides or small molecule inhibitors of the membrane fusion process act to inhibit the conformational changes in S2 associated with fusion or directly prevent the formation of the stabilized S2 postfusion conformation.

Following receptor binding and furin cleavage, further conformational changes in the S2 subunit occur that facilitate membrane fusion. The first of these is that the HR1 undergoes a jack-knife refolding change that allows the fusion protein insertion into the host cell membrane [29,30]. In the metastable prefusion conformation, the HR1 motif folds into four individual α-helix segments that pack against various structural elements, including the long structural 3-helix bundle formed by the CH domain. In the postfusion conformation, the HR1 motif refolds into a long central α-helical coiled coil, as shown in Figure 1C, and forms a 6-helix bundle (6HB) with highly conserved residues from the HR2 motif [29,30]. The extended-stem HR2 element packs against the long central HR1 coiled coil, which induces the binding of the stem helix to the outer region of the central helix, and HR2 into the HR1 binding groove. This serves to strengthen the long central helical bundle formed by SH-HR2 with CH-HR1, and this structure becomes the fusion core in S2, which provides enough stability to the membrane and forms the fusion pore in the postfusion state [30,35].

## 4. Small Molecule Entry Inhibitors of SARS-CoV-2 Spike

Early in the pandemic, numerous drug-repurposing efforts performed screening for antiviral activity using various experimental strategies given what was possible for a novel biological safety level 3 (BSL-3) virus. Since assays using pseudotyped viral particles displaying the Spike protein (and not the viral genome) may be performed in BSL-2 facilities, these assays are much more amenable to high-throughput screening (HTS) efforts. In one of the first HTS drug-repurposing efforts reported (October 2020) using a viral entry assay with pseudotyped particles [36], Chen et al. identified seven Spike-dependent entry inhibitors, where entry inhibitor activity was also measured in multiple cell types (Vero E6, Huh7, and Calu-3 cells). For these entry inhibitors, when antiviral activity was also assessed in cytopathic effect (CPE) assays in Vero E6 cells, it was interesting that very few of the entry inhibitors were found to be full inhibitors with 100% efficacy. In CPE assays, Cepharanthine was able to demonstrate 92.5% efficacy, compared to Abemaciclib (68.7%) and four others that were partial inhibitors: Osimertinib (60%), Trimipramine (48%), NKH477 (45.6%), and Ingenol (38.2%). Of these, Cepharanthine and Abemaciclib exhibited the greatest efficacy and the most robust coronavirus entry activity in multiple cell types (Vero E6, Huh7, and Calu-3 cells) [36]. Given the diverse structural classes of these drugs, the data revealed that it was difficult to identify lower molecular weight (MW) small molecules with 100% entry inhibitor activity. From the structural and medicinal chemistry point of view, the assay data suggested that deciphering the structural requirements of the various compound’s activity would be quite interesting. Why are some compounds full inhibitors and others are not? Can this be explained or rationalized by: (1) differences in 2D structural features on a compound, (2) differences in binding sites or mechanism of action, or (3) differences in the strength of binding ΔG_bind_ at the same binding site? From our point of view, answers to these questions will increasingly reveal the design principles for next-generation inhibitors with improved efficacy. Other subsequent studies and inhibitor screens have been performed with similar pseudotyped particle entry inhibition assays [37,38,39,40], demonstrating a wide range of scientific applications.

While the observed entry inhibitor activity is “Spike-dependent” in a pseudotyped virus assay, the small molecule drugs obviously have other known mechanisms of action. However, entry inhibitor activity may or may not necessarily be related to previously known “host-dependent” mechanisms of action. While numerous viral entry inhibitors have been identified that target a host protein and exhibit “host-dependent” mechanisms of action, we do not aim to summarize those that have been recently reviewed elsewhere [41,42]. Rather, we aim to focus on small molecules that act by binding directly to the Spike protein with various “Spike-dependent” mechanisms of action. Other types of experimental approaches, including biophysical binding studies, mutagenesis, and experimental CryoEM or X-ray structure determination, can characterize more precisely how these small molecules bind to Spike and inhibit viral entry. We also highlight the role of molecular modeling approaches in predicting the location of entry inhibitor binding and the effects on Spike structure and dynamics.

As viral entry is dependent on the structure and function of the trimeric Spike protein expressed on the surface of a mature virus particle, it is not surprising that many viral entry inhibitors target the S1 subunit and disrupt receptor-binding function. Alternatively, we highlight viral entry inhibitors that have been shown to target the S2 subunit and inhibit S2 conformational changes related to S2 function and viral membrane fusion.

### 4.1. S1-Targeted Small Molecule Entry Inhibitors

As the S1 subunit function is to bind to the host (ACE2) receptor through the Spike receptor-binding domain (RBD), numerous immunoglobulins, peptides, and small molecules have been identified that putatively interact with the RBD of Spike [43]. Among numerous research achievements in this area, the designed miniprotein inhibitor from David Baker and colleagues (October 2020) Cao et al. stands out as an impressive milestone in computational molecular design [44]. While we do not aim to review all of these efforts here, we highlight three major small-molecule binding sites in (Figure 2) that are located on the S1 subunit. Numerous small molecules have been reported to bind in the proximity of Site 1, which is found on the RBD in the proximity of res: F342, F374, and F377. As the majority of protein–ligand interactions formed in Site 3 involve RBD residues (res: 390, 517, 519, 546, 565), small molecules that bind to either Site 1 or Site 3 are both considered to bind to the RBD. As much effort has been focused on these sites, several structural classes of small molecules that bind to the RBD have been identified and are reviewed in detail by Sabbah et al. [45]. Experimental complexes of the ACE2 receptor and Spike RBD interactions have revealed complementary electrostatic interactions that facilitate binding at neutral pH, where some ACE2 binding residues are negatively charged, and specific Spike RBD binding residues are positively charged [32]. Thus, it is reasonable to expect that some small-molecule entry inhibitors identified in screening assays may disrupt these interactions by binding directly to ACE2 or the Spike RBD [32] in order to inhibit S1 function.

As the SARS-CoV-2 Spike protein is large and dynamic, until recently, there had been very little experimental structural information on the SAR CoV-2 Spike protein bound to small molecules. Here, we review major proposed small-molecule binding sites on the Spike prefusion conformation (Figure 2) that have not yet been confirmed by X-ray or CryoEM complex structures but are supported by molecular docking studies, biophysical studies, or confirmatory mutations (discussed in detail below). Initial CryoEM complexes of Spike were first reported of the RBD in complex with linoleic acid (Figure 2) (Site 1) [46]. Then, the Biliverdin binding site (Site 2) was found on the S1 N-terminal domain (NTD) [47]. Linoleic acid has now been shown in several structures to bind to the S1 RBD in the free fatty acid (FFA) binding site (Site 1) [46]. The experimental determination of a complex between Spike and the small molecule SPC14 is a breakthrough in showing how a small molecule may bind to the FFA binding site (Site 1) [48]. The CryoEM structure of Spike in complex with the molecule SPC14 (8h3e.pdb) shows how a small molecule containing a heptanoic acid moiety may bind to the FFA site and shift the conformational equilibrium to the closed state, affecting receptor-binding function. The Site 1 binding site on the RBD is defined by res: Y365, L368, F374, F377, L513, F515, V524. In contrast, the Biliverdin site is defined by res: W104, N121, V126, R190, F192.

Site 3 is another major RBD site that has not yet been confirmed by experimental structural biology but is strongly supported by molecular modeling, biophysical studies, and virtual screening efforts. The selective serotonin reuptake inhibitor (SSRI) sertraline is a representative entry inhibitor of SARS-CoV-2 that is proposed to bind to Site 3, where Sertraline has been shown to bind to the S1 RBD, according to biophysical binding data [49]. In the proposed binding model from docking, Sertraline is proposed to bind to Site 3 with key protein–ligand interactions on the RBD (res: 390, 517, 519, 546, 565), where small-molecule binding is on an allosteric interface between the S1 structural domains and parts of the S2 segment in close proximity (res: 968–984) and (res: 747–757). Sertraline binding at this RBD/NTD interface site, as proposed, may rationalize the observation that Sertraline prevented the dissociation of the S1 subunit from the S2 protein [49]. Several groups have modeled basic amine-containing compounds at this site, including chloroquine and derivatives binding to the RBD [50]. Site 3 may also be a possible binding site for other basic amines identified as entry inhibitors, including Trimipramine [36]. Other examples of repurposed antivirals that bind to the RBD include Tilorone and Pyronaridine, which were shown to bind to the RBD by biophysical microscale thermophoresis (MST) experiments [51]. Interestingly, in the same study, another basic amine with a somewhat similar structure, Quinacrine, was not found to bind to the RBD by MST but retained entry inhibitor activity [51]. The most recent structural information for other RBD-interacting entry inhibitors was a recently published series of thiophenyl tryptophan-based trimers that were found to interact with some RBD contacts in an experimental CryoEM structure [52].

### 4.2. S2-Targeted Fusion Inhibitors That Interfere with HR1-HR2 Bundle Assembly

From previous knowledge and research on trimeric viral class I fusion proteins, particularly HIV and influenza, the strategy of inhibiting Spike-mediated viral fusion with peptides that interfere with postfusion assembly has been established. For HIV-1, cleavage results in an N-terminal gp120 containing the RBD and the C-terminal fusion gp41 fragment. For HIV-1, the postfusion 6-helix bundle (6HB) of gp41 is formed from interactions between the heptad regions NHR and CHR. A body of research has shown that peptides based on these NHR and CHR regions are able to prevent 6HB formation [53], but CHR-derived peptides typically have better solubility and often are more potent fusion inhibitors [53].

Following the same rationale, for SARS-CoV and SARS-CoV-2, in the S2 subunit, the equivalent heptad regions HR1 and HR2 form the postfusion 6HB (Figure 1C). Prior to the SARS-CoV-2 outbreak, it had been established that HR2-mimicking peptides of SARS-CoV were effective fusion inhibitors. Subsequently, HR2-based peptides, such as EK1 (March 2020) [37,54], are proposed to bind to the HR1 grove in intermediate states of S2 conformational changes prior to the formation of the final low-energy post-fusion conformation. Peptides such as the 5-HB protein are proposed to bind to the HR2 structural element in the prefusion conformation prior to S2 conformational changes [55,56,57]. Recent advances in peptide-based fusion inhibitors based on HR1 or HR2 sequence peptides have been extensively reviewed recently [58]. We continue our focus on small-molecule fusion inhibitors in the next section, starting with small-molecule fusion inhibitors that mimic the action of these HR1-targeted peptides and interfere with 6HB formation.

### 4.3. S2-Targeted Small Molecules That Interfere with HR1-HR2 Bundle Assembly

One of the first reported (August 2020) small molecule fusion inhibitors of SARS-CoV-2 was salvianolic acid C [59], which is a natural product derived from the traditional Chinese medicine (TCM) product Danshen. Salvianolic acid C is a small-molecule natural product inhibitor proposed to bind to the HR1 grove and prevent assembly of the 6-helix bundle (6HB) postfusion core [59]. Using native-polyacrylamide gel electrophoresis (N-PAGE) and biophysical circular-dichroism (CD) experiments with HR1 and HR2 peptides, salvianolic acid C was shown to interfere with the formation of the 6HB fusion core. Docking studies show that salvianolic acid C likely binds to HR1 residues (940–949), interfering with HR2 peptide docking to the HR1 postfusion core grove. Thus, the mechanism of action of salvianolic acid C is similar to HR2-based peptides, such as EK1, that bind to the HR1 groove as anti-HR1 inhibitors.

Recently, posaconazole was identified as a potent inhibitor and has been proposed to bind to the E1182-L1186-L1193 motif on the HR2 domain and prevent assembly of the 6HB postfusion core [60]. Using mutations, R1185A and N1192A from the HR2 fusion core was shown to reduce the inhibitory effects of posaconazole in pseudotyped virus entry assays. Structurally-related itraconazole [61] was previously identified as a fusion inhibitor of the 6HB post-fusion core [61]. Navitoclax has also been identified as a fusion inhibitor that is reported to target HR1 in the S2 segment [62]. Using an N-PAGE gel-based assay, the authors concluded that navitoclax bound to HR1 peptides and interfered with HR1/HR2 assembly of the 6HB postfusion core. Navitoclax has also been demonstrated to exhibit broad-spectrum activity against a range of VOCs but also retains some activity against SARS-CoV, MERS-CoV, and HCoV-299E [62]. Numerous studies of both peptides and small molecules that target disrupting the 6HB have demonstrated promising broad-spectrum activity against sequence variants or more distantly related coronaviruses [63,64]. These results highlight the advantages of targeting the HR1/HR2 assembly of the 6HB postfusion core.

### 4.4. S2-Targeted Small Molecules That Bind to the S2 Prefusion Conformation

The small-molecule influenza HA-inhibitor arbidol was one of the earliest (May 2020) identified SARS-CoV-2 entry inhibitors [64]. The antiviral activity of arbidol was subsequently verified [65] and published in other studies [66,67]. Our laboratory was very interested in identifying thermodynamically favorable small-molecule binding sites on the S2 subunit and published (February 2021) a map of the TOP50 most favorable binding sites on S2 [15] and compared the results to the crystal structure of arbidol bound to influenza HA2 [15]. Prior to our study, the first report (August 2020) to successfully predict the binding site of arbidol on the SARS-CoV-2 S2 subunit was a computational docking study of Vankadari, where molecular docking was performed on the Spike monomer alone [68]. Nevertheless, the prediction of the binding site location was ultimately correct. Our laboratory also identified and predicted the same binding site [15,69] for arbidol on S2 trimers (December 2021), using molecular docking and structure–activity relationship (SAR) modeling of the experimental activity of derivatives [69]. A nearly simultaneous (December 2021) publication from Shuster et al. [70] experimentally confirmed the binding site of arbidol on the S2 segment of Spike. Shuster et al. [70] identified the site experimentally using a chemical biology approach and then corroborated the exact predicted binding site [15,68,69] by mutational studies [70]. Thus, while the location of the arbidol binding site on the S2 segment had been unclear to that point, simultaneous publications from two laboratories [69,70] identified and independently confirmed the site initially reported by Vankadari [68] using entirely different approaches. Thus, arbidol is the best representative of small-molecule fusion inhibitors that bind to Site 5 on the S2 segment, as shown in Figure 2A. In a chemical series of oleanolic acid (OA) saponin derivative fusion inhibitors, the representative compound **12f** was shown to bind to the S2 segment by surface plasmon resonance (SPR) [71]. Subsequent molecular modeling studies of the structure–activity relationship (SAR) suggested that the series of OA saponin derivatives, including **12f,** was best modeled binding to the arbidol binding site [69].

Subsequently, other molecules have been reported to bind at or in proximity to the arbidol site. NBCoV1 and NBCoV2 are small molecules that were shown to bind to the Wuhan lineage Spike trimer by SPR with an affinity of 1.56 μM and 5.37 μM, respectively [72]. However, they were found to bind much weaker to constructs of the S1 segment; thus, the authors concluded that the molecules may bind to S2 [72]. The NBCoV1 and NBCoV2 compounds were tested using a number of mutants from VOCs, B1.b.7 UK triple mutant (d67-70/N501Y/P681H) and B.1.617.2 Delta single and triple mutants in particular. Interestingly, NBCoV2 (WT 22.8 μM) was found to lose the most activity with the WT D950N single mutant (88.7 μM) (from the B.1.617.2 Delta), while the WT D614G mutant (61 μM) and E484K (45 μM) also lost activity. While D950N is the only S2 mutation that was shown to demonstrate the largest shift, other mutations that are either in medium proximity ~27 Angstroms (D614G) or are quite distant from the site, 70–80 Angstroms (E484K), are both able to show significant effects in dose responses, illustrating obvious long-range allosteric effects that have been studied by other structural and biophysical techniques to probe structure and dynamics. In another study, NBCoV63 [73] was identified as a fusion inhibitor that is similar in structure to NBCoV1 in that it contains a 3-(furan-2-yl) benzoic acid substructure. While NBCoV63 exhibited lower potency, it exhibits much better drug-like properties in that it does not contain a functional group implicated as a frequent hitter [74,75]. In the docking study, the NBCoV63 benzoic acid formed a salt bridge with K947 of HR1, which forms a salt bridge with E1182 of HR2 in the postfusion conformation. In the docking model, electrostatic interactions of the NBCoV63 acid (COO-) with K776 and K947 were key to the binding mode, very similar to the binding mode of arbidol. NBCoV63 was also shown to retain activity for Omicron (BA.4/BA.5) variants in pseudovirus assays in both 293T/ACE2 and A549/ACE2/TMPRSS2 cell lines [73]. Next, we will summarize small molecules that are proposed to bind at other sites on S2.

### 4.5. Other Proposed Fusion Inhibitor Binding Sites on the S2 Segment

Site 4 is found on the interface of S1 and S2, as shown in (Figure 3), and at least five different entry inhibitors have been modeled there by various groups. Nelfinavir was one of the first (May 2020) small molecules that were reported to bind at this site, according to molecular docking [76]. Subsequently (September 2020), other entry inhibitors, such as toremifene, have also been proposed to bind at Site 4 according to molecular docking [77]. Interestingly, prior to the pandemic, it was known that some natural product saponin derivatives (glycyrrhizin, aescin, α-hederin) exhibited entry inhibitor activity [78], and several new series of saponin derivative natural products have been shown to be SARS-CoV-2 inhibitors [79,80]. As discussed previously, OA saponin derivative **12f** was found to bind to the S2 subunit by SPR [71]. Similarly structured ursolic acid derivatives, such as **UA-30,** have also been reported to bind at the Site 4 S1/S2 interface site [79]. In a new series of 3-O-b-chacotriosyl ursolic acid derivatives, the derivative **UA-18** was shown to bind to Spike by SPR and is shown to be best modeled as binding to this S1 and S2 interface site [79]. In this study, the double mutants (N764A/R765A and Q957A/K964A) and quadruple mutant (N764A/R765A/Q957A/K964A) [79] were used to confirm the **UA-18** binding site. Interestingly, in another study of betulonic acid saponin derivatives, **BA-4** is a representative molecule that shows strong binding to S2 by SPR and does not bind to S1 [80]. Shown in Figure 3 is an example model of OA saponins binding this site, which is very similar to published models [79,80]. However, molecular modeling studies from our laboratory have also concluded that at some specific OA, saponins, such as **12a** and **12f**, are predicted to bind more favorably at the arbidol Site 5 rather than Site 4 [69]. In summary, toremifene, nelfinavir, and a variety of natural product saponin derivatives are proposed to bind at Site 4, and presumably, small-molecule binding at this site acts to stabilize the prefusion conformation.

Moving our attention to another proposed binding site, as shown in Figure 4, Site 6 on the S2 segment of the SARS-CoV-2 Spike is analogous to the proposed camphecene binding site on Influenza HA [81], which was confirmed by mutagenesis and resistance mutations [82,83]. For a novel series of borneol ester derivatives, Yarovaya et al. identified that derivatives **11** and **21** had the greatest activity in pseudovirus infections in both original Wuhan and Delta (B.1.617.2) strains [84]. The authors conclude that derivatives **11** and **21** are well-modeled binding to an allosteric site that is analogous to the camphecene binding site on influenza HA [81]. On the SARS-CoV-2 Spike protein, this binding site for borneol esters **11** and **12** is in close proximity to the fusion peptide helical conformation binding site in the prefusion conformation of Spike [84]. As shown in Figure 4C key protein–ligand interactions that define this site using Wuhan strain numbering include S2 residues that are just C-terminal of the fusion peptide (F823, L826, and F833) and residues (P1057, H1058) from the S2 connector domain (CD) (res: 986–1035). The borneol ester derivative binding site on the SARS-CoV-2 Spike is supported by sequence analysis, extensive molecular docking, and molecular dynamic (MD) simulations and supported by the rationale of how camphecene derivatives are proposed to bind to influenza HA [81,82,83].

Last, we would also like to highlight Site 7 shown in Figure 4, an additional putative binding site on the S2 segment for both peptides and small molecules. Site 7 on the S2 subunit has been reported to be thermodynamically favorable for binding small molecules in several reports [15,69,85,86]. In this binding site, a key hydrophobic contact from residue W886 is an important structural element of a long helix-turn-helix (res: 867–890), which is N-terminal to the HR1 and undergoes major conformational changes upon the transition from the prefusion to the postfusion conformation. In the prefusion conformation, this site is formed from W886, four residues from the N-terminus of the CD (Q1036, K1038, V1040, Y1047), and residues R1107 and N1108, which are N-terminal to the SH domain also undergo major refolding in the prefusion to postfusion transition. To our knowledge, the thermodynamic favorability for this site was first reported from a virtual screening study that was focused on the S2 segment that identified chitosan to bind in proximity to this site [85]. In our study using a pharmacophore mapping, docking, and free energy approach, we identified this specific site as one of the most thermodynamically favorable binding sites on the S2 segment in the prefusion conformation [15,69] as well as the arbidol site (discussed above). In another study, a fusion inhibitor tripeptide VFI was also proposed to dock to this site, shown in (Figure 4D), based on docking the VFI tripeptide to the entire Spike protein [86]. While this site has not yet been validated by mutagenesis, our laboratory, and others will aim to see if point mutants affect fusion or small-molecule binding. With recent advances in CryoEM characterization of Spike conformational states, it may be possible to characterize small molecule complexes of representative fusion inhibitors bound to Spike. As with other viruses in the past, such as Influenza and HIV, prior to small-molecule binding sites being experimentally confirmed by either mutagenesis or X-ray crystallography, it is important to remember that there is an important role for molecular modeling approaches to predict likely binding sites and to guide experimental approaches towards confirming them.

### 4.6. Potential Advantages of S2-Targeted Antivirals

One potential advantage of S2-targeted peptides and small molecule antivirals is that, based on sequencing, they may be expected to exhibit a broader spectrum of activity for coronaviruses that are more distantly related in sequence. In general, the key amino acids that form the binding sites on the S2 subunit are much more conserved than sites on S1. These sites may offer the opportunity to develop agents that exhibit either a broader spectrum or a higher barrier to antiviral resistance. Recent studies have uncovered some potential advantages of S2-targeted immunoglobulins [25]. As mentioned previously, not only peptides but also small molecules targeting the HR1/HR2 assembly of the 6HB postfusion core exhibit some level of broad-spectrum activity. Posaconazole [60] and navitoclax [62] demonstrated broad-spectrum activity against a range of VOCs. In addition, Novitoclax was also found to have activity against SARS-CoV, MERS-CoV, and HCoV-299E [62].

Beyond targeting HR1/HR2 assembly, interestingly, S2-targeted small molecules proposed to bind to the prefusion conformation of Spike at Site 4, 5, 6, and 7, which all demonstrate some evidence of broad-spectrum antiviral activity. Small molecules proposed to bind at Site 4 have been shown to have some broad-spectrum activity, including various saponin derivatives with activity for several VOCs [79,80]. Arbidol has been shown to bind to Site 5 and exhibit broad-spectrum activity, particularly in that it had been known to bind to Influenza HA [68,69]. Similarly, the small molecule NBCoV63 is also proposed to bind at the same Site 5 on Spike and retains activity for Omicron (BA.4/BA.5) variants [73]. Borneol Ester derivatives proposed to bind at Site 6 on Spike also demonstrate broad-spectrum activity, where some derivatives were known to bind to Influenza HA but also inhibit SARS-CoV-2 Wuhan lineage B, Delta (B.1.617.2), and Omicron (B.1.1.529) [84]. Lastly, the inhibitory VFI peptide that was identified and proposed to bind at S2 site 7 also exhibited antiviral activity for both SARS-CoV-2 and hCoV-OC43 [86] and demonstrated greater antiviral effects in pre-treatment assays like other fusion inhibitors.

In summary, S2-targeted small molecules from various structural classes have shown promising activity and act through a range of proposed mechanisms of action. These include inhibiting the HR1/HR2 assembly of the 6HB postfusion core as well as binding to Spike in the prefusion conformation at Sites 4, 5, 6, or 7 to prevent conformational changes in S2 structure that are required for S2 function.

## 5. Conclusions

We reviewed a range of small molecules that act by binding directly to the Spike protein with various “Spike-dependent” mechanisms of action. While numerous viral entry inhibitors targeting the S1 subunit have been identified, we highlighted small molecules targeting the S2 subunit. S2-targeted peptides that interfere with HR1-HR2 bundle assembly are very promising and have been shown to be effective and exhibit broad-spectrum activity. Small molecules such as salvianolic acid C, posaconazole, itraconazole, and navitoclax act using a similar mechanism of action to block the HR1/HR2 postfusion 6HB assembly. We summarized seven proposed small-molecule binding sites on the SARS-CoV-2 Spike protein in the prefusion conformation. While Site 1 and Site 2 on the S1 subunit are confirmed by experimental CryoEM structure determination, Site 4 and Site 5 involving the S2 subunit are strongly supported by experimental point mutations and more than one independent study identifying the same binding site. Site 4 on the allosteric interface of the S1 and the S2 segment is where reference inhibitors nelfinavir, toremifene, and natural product saponin derivatives have been proposed to bind. Site 5 is found on the S2 subunit where arbidol is proposed to bind [68,69], as well as other small molecules such as NBCoV63 [73]. Small molecule binding at Site 4 and Site 5 on the S2 subunit are proposed to inhibit S2 conformational changes related to S2 function and viral membrane fusion. Future efforts to confirm the binding site location of representative small molecule inhibitors will elucidate how specific small molecule inhibitors inhibit S2 conformational changes. Future mutagenesis efforts interrogating these sites (pseudovirus fusion assay activity, biophysical small molecule binding, biophysical structure, and function studies) should be able to delineate how small molecule binding prevents conformational changes in S2. Efforts in this direction may reveal new attractive strategies to target S2 and may also identify previously unknown functions of these binding sites. While significant attention has gone to therapeutic strategies focused on targeting S1, we highlighted the potential advantages of S2-targeted peptide and small-molecule inhibitors.

## Figures and Tables

**Figure 1 viruses-16-00712-f001:**
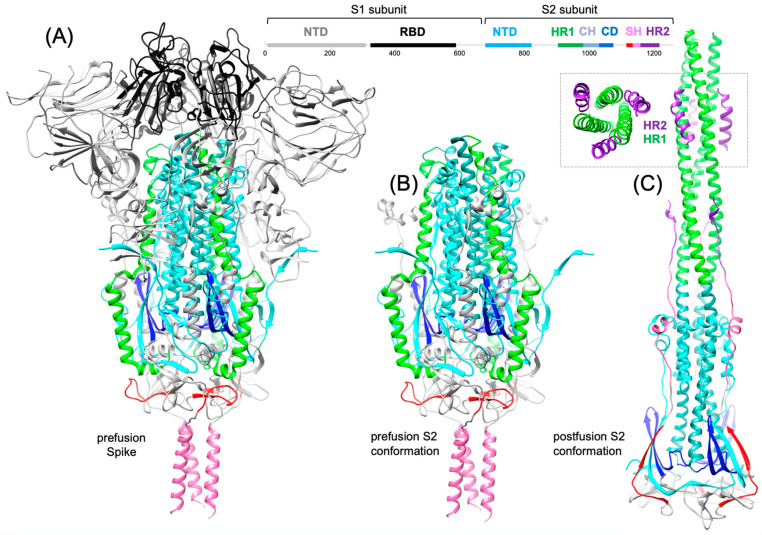
Structure of the SARS-CoV-2 Spike protein showing: (**A**) the prefusion conformation (6xr8.pdb) full-length trimeric protein, (**B**) the prefusion conformation of the S2 subunit, and (**C**) the postfusion conformation (6xra.pdb) of the S2 subunit showing the heptad repeats HR1/HR2 6-helix bundle. On Spike, residues are colored gray S1 N-terminal domain (NTD) (res: 14–304), black S1 receptor-binding domain (RBD) (res: 331–528), cyan S2 NTD (res: 685–816), green HR1 (res: 910–985), seafoam green central helix (CH) (res: 985–1035), dark blue connector domain (CD) (res: 1035–1068), red (res: 1112–1134), stem helix (SH) pink (res: 1140–1161) and HR2 (res: 1163–1211).

**Figure 2 viruses-16-00712-f002:**
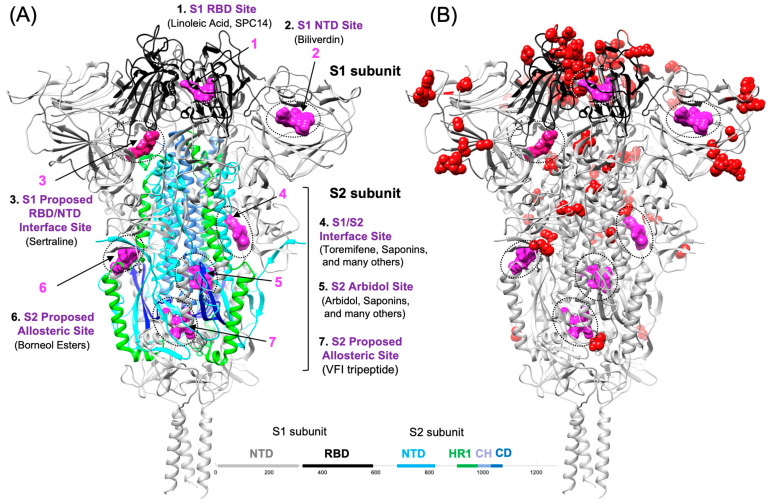
Small-molecule binding sites on the SARS-CoV-2 Spike protein. (**A**) Seven proposed or experimentally determined small-molecule binding sites are shown as magenta (molecular surfaces) on the ribbon structure of the full-length trimeric protein (6xr8.pdb). (**B**) On Spike, small-molecule binding sites are shown in magenta, and residue positions implicated in major SARS-CoV-2 VOCs are shown in red. Most substitutions in the Spike protein in VOCs to date have been found in the S1 segment.

**Figure 3 viruses-16-00712-f003:**
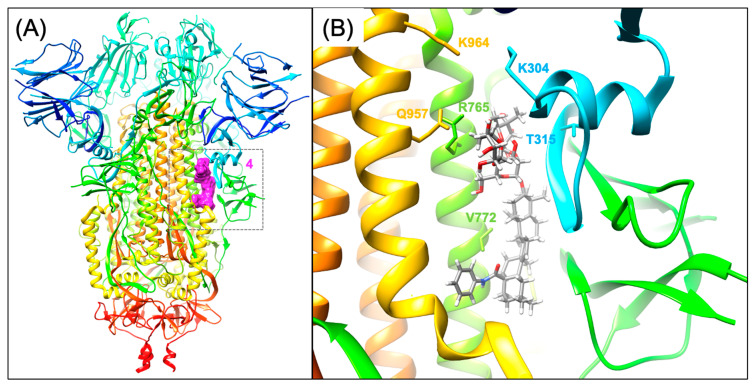
Small molecule binding site 4 on the S1/S2 interface. (**A**) Molecular surface of modeled oleanolic acid (OA) saponin **12a** shown in magenta binding to Spike (6xr8.pdb) colored rainbow from blue (N-term) to red (C-term). (**B**) Model of OA saponin **12a** binding to site 4.

**Figure 4 viruses-16-00712-f004:**
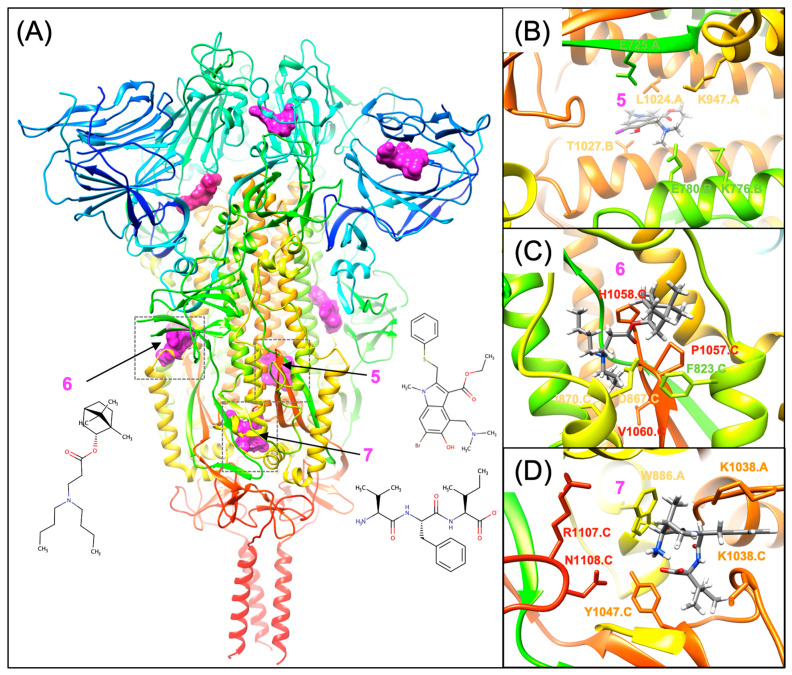
Other small molecule binding sites on S2 subunit. (**A**) Molecular surface of small molecules shown in magenta binding to Spike (6xr8.pdb) colored rainbow from blue (N-term) to red (C-term). Important binding site residues are shown using the same residue coloring for (**B**) arbidol binding to site 5. (**C**) Borneol Ester **21** binding to site 6 and (**D**) tripeptide VFI binding to site 7.

## Data Availability

All relevant data are shown in figures or included in the text.

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
