# Peer review of "Research Progress on Spike-Dependent SARS-CoV-2 Fusion Inhibitors and Small Molecules Targeting the S2 Subunit of Spike"

_viruses, 2024, doi:10.3390/v16050712_

Round 1

Reviewer 1 Report

Comments and Suggestions for Authors

Review report for viruses-2920997

The manuscript (viruses-2920997), “Research Progress on Spike-Dependent SARS CoV-2 fusion inhibitors and small molecules targeting the S2 subunit of Spike” authored by Matthew R. Freidel and Roger S. Armen summarize the current status of drug re-purposing efforts in developing and identifying new peptides and small molecules of viral entry and fusion inhibitors that directly bind to the SARS CoV-2 Spike protein, particularly  focus on those bind to the S2 subunit. The review has cited a relatively complete publications and is well written, thus provide useful information for researchers in this field.

Some minor comments/suggestions:

P7, line 235: “S2-tarteted” >> “S2-targeted”

P7, line 278: “they authors” should be “the authors”.

P8, line 310: what is the OA? Authors should give the full name when use an abbreviation first time in the text. Such as SPR on P8, line 311, MD on P10, line 386, et al.

P10, lines 372-375: please re-write this sentence for clarity with proper language.

Comments on the Quality of English Language

Need minor editing.

Reviewer 2 Report

Comments and Suggestions for Authors

In the present manuscript, the authors describe a variety of peptides and small molecule inhibitors that bind Spike protein, focusing on the S2 subunit. The aim to review research progresses on small molecule viral entry and fusion inhibitors that directly bind SARS-CoV-2 Spike protein is achieved. 

The text is well organized in different paragraphs that separate the identification and the mechanisms of action of the various inhibitor compounds.

Main comments:

-          The paragraph 3 ‘Conformational changes and S2 subunit function’ and all the text in general, presents the main conformational changes of Spike (specifically S2). I strongly invite the authors to insert a cartoon that let to follow all these key steps.

-          In the paragraph 4 ‘Small molecule entry inhibitors of SARS-CoV-2 Spike’ the authors present only the reference [32] to show the entry inhibition of pseudoviral particles. Please, add other references possible published in the last two years.

-          I encourage the authors to insert in the Figure 3 ‘Small molecule binding site 4 on the S1/S2 interface’ other cartoons similar to b) that show the binding of inhibithors to all the putative sites on S2 (site 5, 6 and 7).

-          The references presented are very impressive. Nevertheless I suggest not to repeat the same reference in the same sentence (as example, see line 275-276 for the reference [54]).    

Line 278: Replace ‘they’ with ‘the authors’

Line 317: Not clear which is the Spike variant that shows 1.56 µM and 5.37 µM affinity values. UK? Delta? Wuhan?

Line 322: Replace ‘NBoV2’ with ‘NBCoV2’

Please verify in the text that the molecule inhibitors presented are all with the capital initial.

Reviewer 3 Report

Comments and Suggestions for Authors

This review focuses on small molecules, which could be useful because small molecules, whether in the existing form or derived from, are relatively easy to mass-produce and to get through regulatory authorities. However, the review does not address any key issues as I list below.

First, as the authors stated, the small molecules inhibit the viral entry of host cells mainly through binding to the S1 or S2 domain. The initial binding can occur in two ways. The first  is through electrostatic interactions (PMID: 29319301). For example, under normal extracellular pH, the distal surface of the receptor-binding motif is positively charged, and that of the ACE2 is negatively charged, so that two are electrostatically attractive to each other (PMID: 37560522). The second is through interaction between uncharged domains in the SARS-2S and uncharged small molecules. There are many structures of SARS-2S, one can use docking experiments or dynamic modelling to identify binding locations of the small molecules. The review reads more like an assembly of abstracts rather than a critical analysis.

The review is supposed to integrate recent progress on small molecules inhibiting the membrane fusion. There are two pathways of membrane fusion, one between the viral membrane and the host cell membrane, and the other between the membrane of an infected cell and a neighboring uninfected cell to form syncytium to facilitate viral spread from the former to the latter (PMID: 32362314, 33466921). SARS-CoV uses mostly the first, and SARS-CoV-2 uses mostly the second. Which pathway to take depends on how the spike protein is cleaved. The two references (and the literature cited therein) contains detailed information. Which pathway are these small molecules targeting? The review is a bit too superficial.

Discussion of drug efficacy is always associated with potential side effects. For example, the SARS-S synthesis takes place in the lumen of endoplasmic reticulum, and then modified progressively when it is exported through Golgi apparatus (PMID: 19258633, 15367599). This most likely is also true for SARS-2S. Misfolding of the spike protein can cause an unfolded protein response (UPR). Will the small molecules cause structural changes in the spike protein and trigger UPR? They could also cause many other side effects. Structural approaches to such problems are to identify which peptide motifs these small molecules bind to, and then check what other human proteins, especially those important ones, also share such motifs.

There are many experimental and bioinformatic approaches to identify COVID drugs with the greatest efficacy and smallest side effects. Some approaches lead to significant insights while others not. A review should critically evaluate these approaches to help readers to understand the differences and effectiveness of these approaches.

The authors appear very fond of the word "numerous". It is always better to replace it with an actual number.

The authors often confuse mutation with substitution. For example, through statements such as "The S1 segment is highly subject to mutation", they often seem to think that S1 experiences more mutations than S2. Mutations do not occur preferentially on S1 relative to S2. However, selection tend to filter out deleterious mutations and preserve beneficial ones. Many mutations in S1 confers some evolutionary benefits and are preserved by natural selection, not because "The S1 segment is highly subject to mutation".

Round 2

Reviewer 3 Report

Comments and Suggestions for Authors

The authors misunderstood what I said from the very beginning. They stated in the manuscript that the small molecules inhibit the viral entry of host cells mainly through binding to the S1 or S2 domain, so I asked them to be more specific on what types of binding they were referring to. To help them answer my question, I outlined two possible types of binding: 1) binding between two partners carrying opposite electric charges, and 2) binding between two uncharged partners. What types of binding were they referring to? The electrostatic interactions are often nonspecific but facilitate subsequent more specific bindiing. What have other researchers suggested? What is the authors' opinion on what others have suggested?

The authors did not answer the question as they apparently did not take the trouble to understand the question. In fact, they broke 1) and 2) as if they were two unrelated questions.

Without anything specific on binding/interactions between small molecules and SARS-2S, how would one know that the small moecules are targeting S2 instead of S1?

The authors’ responses to some other questions also represent misunderstandings.
